# Role of Cardiovascular Imaging in the Follow-Up of Patients with Fontan Circulation

**DOI:** 10.3390/children9121875

**Published:** 2022-11-30

**Authors:** Sara Moscatelli, Nunzia Borrelli, Jolanda Sabatino, Isabella Leo, Martina Avesani, Claudia Montanaro, Giovanni Di Salvo

**Affiliations:** 1Paediatric Cardiology Department, Royal Brompton Hospital Harefield NHS Foundation Trust, London SW3 5NP, UK; 2Adult Congenital Heart Disease Unit, A.O. dei Colli, Monaldi Hospital, 80131 Naples, Italy; 3Division of Paediatric Cardiology, Department of Women and Children’s Health, University Hospital Padua, 35128 Padua, Italy; 4Paediatric Research Institute (IRP), Città Della Speranza, 35127 Padua, Italy; 5Department of Medical and Surgical Sciences, Magna Grecia University, 88100 Catanzaro, Italy; 6Cardiac Magnetic Resonance Department, Royal Brompton Hospital Harefield NHS Foundation Trust, London SW3 5NP, UK; 7Adult Congenital Department, Royal Brompton Hospital & Harefield NHS Foundation Trust, London SW3 5NP, UK

**Keywords:** Fontan, TCPC, cardiovascular multimodality imaging

## Abstract

Since its first description in 1971, the Fontan procedure and its modifications have led to a substantial improvement in the survival rates of patients with a variety of types of complex Congenital Heart Disease (CHD) characterised by the presence of a single, dominant ventricle. However, despite the significant improvement of the prognosis over the years, Fontan patients are still exposed to several cardiovascular and systemic complications. It is, therefore, important to fully understand the pitfalls hidden behind a Fontan anatomy and the potential predictors of ventricular failure. Cardiovascular imaging plays a key role in this context, allowing for the early identification of complications with important prognostic implications. Echocardiography remains the first-line imaging modality for serial evaluation of Fontan patients. However, there is a growing role of cardiovascular magnetic resonance and cardiac computed tomography from pre-operative assessment to longitudinal follow-up. The aim of this paper will be to provide a comprehensive overview of the role, strengths, and weaknesses of each imaging modality in the assessment of congenital cardiac conditions palliated with the Fontan procedure.

## 1. Introduction

Children as well as adults with a single ventricle (SV) after Fontan palliation are exposed to numerous complications over the course of their lifetimes, such as failure of the SV performance [1,2]. The modern version of the Fontan circuit is the total cavopulmonary circulation (TCPC) [3] in which the pulmonary circulation is positioned, like a “dam”, between the systemic venous return and the systemic ventricle with both of its most recognised characteristics: upstream congestion and downstream decreased flow. Then, by creating the Fontan circuit, the “dam” controls the overall systemic flow and the degree of congestion upstream, and the critical bottleneck is, therefore, shifted outside and upstream of the heart itself [3]. Numerous factors are currently believed to play a detrimental effect on ventricular function in the Fontan population, which may lead, in turn, to cardiac transplantation and/or death [1,2].

Recognizing predictors of ventricular failure is an unmet need of current clinical research, in order to identify patients at a higher risk who need closer monitoring, along with potential percutaneous or surgical interventions to prevent or relieve ventricular failure [1,2,3,4].

“A big ventricle is a bad ventricle” is what, as cardiologists, we have learned for decades. On the contrary, when referring to Fontan patients, it is known that there is a short route between an underfilled ventricle and one with a volume overload, because the SV is acutely sensitive to a preload reduction, which would obligate the patient into a low-output state [1,2,3,4,5].

Single ventricles may be, indeed, relatively dilated in order to perform, on the Starling Curve, at increased contractility, considering that an SV with a Fontan circulation is required to work as two ventricles. However, just as for normal hearts, there is a break point of dilatation that may become damaging [1,2,3,4,5]. 

Many underling reasons may explain what usually happens besides an intrinsic myocardial failure, such as the atrio-ventricular and semilunar valve regurgitation, the chronotropic insufficiency, and the systemic to pulmonary collateral burden [1,2,3,4,5]. Furthermore, not only myocardial fibrosis, as an obvious trigger for ventricular dilatation and heart failure, but also hepatic fibrosis can play a pivotal role in this process. As a matter of fact, when hepatic fibrosis increases, there is a rising of the inferior vena cava flow, due reasonably, in part, to hepatic arterialisation [6]. 

In these circumstances, the (single) ventricular work and output are increased as well. Accordingly, hepatic fibrosis, which is a common complication, may cause an additional volume overload in an already compromised ventricle, potentially accentuating cardiac fibrosis and ventricular failure along with a poor outcome [6].

Assessing the ventricular end-diastolic volume may be complicated in Fontan patients, as the heterogeneity of the anatomy cannot be compared in a unique cohort (mostly single chamber or unbalanced two chambers, right ventricle (RV) or left ventricle (LV) morphology, D-looped or L-looped).

For these reasons, cardiovascular magnetic resonance (CMR) is the gold standard for assessing ventricular volumes and performance according to the most valuable recommendations [4].

Most recently, speckle-tracking echocardiography (STE) has been confirmed as being a reliable technique, allowing for the evaluation of ventricular deformation without any geometric assumption or the impact of acute preload changes [7]. Although some recent studies have employed STE to measure ventricular performance in Fontan patients, just a few have succeeded in demonstrating its correlation with adverse prognosis.

Borrelli et al. [8] recently demonstrated that an impairment of longitudinal strain predicts adverse outcomes in children with hypoplastic left heart syndrome (HLHS). Accordingly, serial measurements of longitudinal strain (LS) are to be encouraged for clinicians who are dealing with Fontan patients, to identify the subgroup of SVs at higher risk. 

Meyer et al. [9] brought attention to the value of integrating strain analysis with the measurement of ventricular volumes by CMR. They have in fact demonstrated, in a cohort of 416 Fontan patients, that increased ventricular dilatation was the strongest independent predictor of death or transplant in the long-term follow-up and that patients with both ventricular dilatation and a worse global circumferential strain (GCS) were at the highest risk.

The scope of this review is to summarise and propose a surveillance imaging testing toolkit for the follow-up care for patients with a Fontan circulation. Gaps in knowledge along with areas for future focus of investigation are emphasised, with the aim of building the best follow-up protocol in order to create a normal life for these unique individuals.

## 2. Echocardiography in Fontan Circulation

Transthoracic echocardiography (TTE) remains the cornerstone technique of cardiac diagnostic imaging in patients with congenital heart disease. Considering the large accessibility, reproducibility, and ease of use of the method, it has been widely adopted even in patients with single-ventricle physiology [10]. However, imaging patients with a Fontan circulation can be particularly challenging considering the complexity of the surgical technique and of the underlying cardiopathy and deterioration of the image quality caused by age and previous surgeries.

### 2.1. Segmental Analysis

When a patient with complex univentricular cardiopathy presents with an unclear surgical history and anatomical details, a comprehensive evaluation is needed to define the anatomical and haemodynamic characteristics of the single-ventricular circulation. It is necessary to use a methodical process for the segmental analysis in order to provide a reliable morphological assessment [11].

The segmental analysis involves the evaluation of anatomical and embryological blocks, which are the main components of the mammalian heart, in a step-by-step approach. The veins and atria represent the first segment, followed by the ventricles and major arteries. The cardiac position, function, and connection of all segments should then be described. First, the location of the heart is defined, along with the anatomical pattern of the abdominal and thoracic organs.

Situs solitus is the usual arrangement of the body’s organs, where the liver, cecum, and appendix are right-sided, and the spleen, pancreas, stomach, and sigmoid colon are left-sided. The left mainstem bronchus is longer and more horizontal and runs inferior to the left pulmonary artery, and the left lung consists of two lobes. The right mainstem bronchus is shorter and runs posterior to the right pulmonary artery, and the right lung has three lobes. In situs viscerum inversus, the orientation of the body organs is reversed, mirroring that of situs solitus. The third arrangement, situs ambiguous (left and right isomerism), is characterised by the abnormal symmetricity of the body organs. Since the echocardiographic evaluation of the atria situs can be difficult, the abdominal and thoracic situs are generally used to guide the definition of the atrial situs. However, it should be noted that there is no 100% correlation between the thoracic or abdominal characteristics of the visceral situs and the atrial situs. The echocardiographic short-axis subcostal view is commonly used to define the visceral situs. In situs solitus, the aorta is pictured as a pulsatile vessel anterior; to the left of the spine, the inferior vena cava (IVC) is pictured as a collapsible vessel further anterior; and, to the right of the spine, the liver is rightward, while the spleen is leftward. This visceral arrangement is generally associated with atrial situs solitus. In situs inversus, the aorta is visualised to the right of the spine, while the IVC is to the left. Accordingly, the right atrium and left atrium are in a mirror position.

Cardiac position refers to the heart localisation relatively to the midline. The terms levoposition and dextroposition denote the majority of heart mass to the left or to the right of the midline, respectively. When the heart is mesopositioned, it is located around the midline.

Cardiac orientation refers, instead, to the orientation of the base-to-apex axis. Normally, the base-to-apex axis points to the left, defining the cardiac levocardia. In dextrocardia and in mesocardiac, the base-to-apex axis points to the right and downward, respectively.

The next step is defining the atrio–ventricular connection and ventricular looping. Ventricular looping is the term used to describe how the heart tube loops during embryological development. When the looping is to the right (D-looping), the right ventricle is anterior and to the right of the left ventricle. When the looping is to the left (L-looping), the morphological right ventricle is to the left of the morphological left ventricle. Then, it should be defined if a biventricular or single-ventricle connection is present. While in a biventricular connection two atrio-ventricular valves are present with two possible connections of the atria to the ventricles (concordant or discordant), a univentricular connection is more challenging.

The term SV refers to various forms of complex congenital heart defects in which there is one single-ventricle chamber or a large dominant ventricle with a rudimentary ventricle.

In the presence of a single functional ventricle, a double inlet (where two atria empty into a single ventricle) or a single inlet (where a single atrium empties into a single ventricle) can be present.

A ventricle morphology evaluation can also be tricky in univentricular hearts. In this case, the type of rudimentary ventricle can give some clues. If the rudimentary ventricle is positioned posteriorly to the main ventricle, the single-ventricle morphology is of the right type. On the contrary, if the rudimentary ventricle is positioned anteriorly to the main ventricle, the single-ventricle morphology is of the left type.

Finally, the ventricular–arterial connection is determined. A concordant, discordant, or double-outlet connection can be present. In a concordant connection, the aorta arises from the left ventricle, while the pulmonary artery arises from the right ventricle. In a discordant connection, on the contrary, the aorta arises from the right ventricle, while the pulmonary artery arises from the right ventricle. If one artery arises completely from the right ventricle, and the other arises more than 50% from the same right ventricle, a double-committed connection is present.

However, in a Fontan circulation, both the main arteries serve as the systemic outflow of the single ventricle [11].

### 2.2. Single Ventricle Function

As ventricular function in the palliated univentricular heart declines, there is a significant potential for negative clinical outcomes. Considering the lack of a subpulmonary ventricular pump, there is a limited possibility to increase the cardiac output by raising the preload of a failing single ventricle. As a consequence, a failing single ventricle of a Fontan circulation results in a low cardiac output status, which is particularly vulnerable.

The dominant ventricle in Fontan patients may be morphologically left, as in tricuspid atresia or a double-inlet left ventricle, or morphologically right, as in hypoplastic left heart syndrome. Moreover, two relatively well-formed ventricles can be present, as in patients with a large ventricle defect where a double-ventricle correction strategy cannot be pursued.

When a morphological left ventricle is present, from the apical four-chamber view, ejection fraction by biplane Simpson’s method is the technique of choice to estimate ventricle function.

More challenging can be the analysis of a morphological right ventricle. From the apical four-chamber view the fractional area change can be evaluated using the formula: [(end-diastolic area − end-systolic area)/end-diastolic area] × 100.

Information on ventricular performance can also be obtained via Doppler-derived systolic rates of ventricular pressure change (dP/dT). It is worth noting, there are no precise reference limits for these functional parameters in single ventricles, so perhaps added value may be gained from follow-up longitudinal reevaluations [12]. 

In adult Fontan patients, Bunting et al. demonstrated that a 3D echocardiography ejection fraction correlated strongly with the one evaluated by CMR and can be used for serial functional assessments [13].

The AV valve S/D ratio allows the assessment of ventricular function without anatomical assumption and represents an independent predictor of mortality in adult patients with a Fontan circulation [14].

Recently, STE has been demonstrated to be a reliable technique to evaluate single-ventricle performance without any geometric assumption or influence of acute preload [15,16]. A drop in GLS after a Norwood procedure has been shown to carry a poor outcome [8,17,18,19]. 

In a study by Rösner et al., a typical dyssynchrony pattern by STE was found in several patients with a Fontan circulation. This pattern was characterised by an early shortening of the dyssynchronous segments (‘flash’), followed by systolic stretching; while, on the contrary, segments with a conduction delay showed early stretching, followed by delayed contraction. Among the different functional parameters, the presence of electromechanical dyssynchrony was the strongest predictor for mortality and heart transplantation [20]. 

Diastolic dysfunction is common in patients with a Fontan circulation, even in the presence of normal systolic function. The significant prolongation (>28 ms) of the pulmonary venous atrial reversal flow, relative to atrial forward flow duration into the ventricle, suggests that the single ventricle’s filling pressure is elevated. Reductions in the deceleration time of the atrio-ventricular valve inflow have been also associated with diastolic dysfunction [21].

Yamazaki et al. demonstrated that Fontan patients performed less well at exercise stress echocardiography when compared to healthy subjects; this was largely due to an impaired contractile reserve, chronotropic incompetence, lower ventricle filling capacity, minimal preload variability, and less afterload reduction [22]. 

The study of fluid dynamics [23,24] can provide more insight in the evaluation of single-ventricle performance. In particular, it has been recently demonstrated that Fontan patients present an increased kinetic energy dissipation with a high vortex flow rotation intensity [24].

Ventricle function assessment by echocardiography is shown in Figure 1.

### 2.3. Atrio-Ventricular Valve Function

High atrial pressure hampers transpulmonary flow, especially in the absence of a subpulmonary ventricle pump, as in a Fontan circulation. Accordingly, as in the case of atrioventricular (AV) valve stenosis or regurgitation, the increase in atrial pressure may be poorly tolerated by Fontan patients. Routine assessment of the AV valve comprises the morphologic evaluation of thickness and mobility of the leaflets, together with a semi-quantitative doppler estimation of AV valve regurgitation (including the vena contracta width, density, and shape of the continuous doppler signal of the regurgitant jet). Any diastolic transvalve gradient should also compromise Fontan physiology and be accurately evaluated.

### 2.4. Venous and Arterial Pathway

Coarctation of the aorta (as in a hypoplastic left heart with a neo-aortic arch), subaortic stenosis (as in a double-inlet left ventricle with a restrictive bulboventricular foramen), and pulmonary arteries distortion are the most frequent issues after the Fontan procedure. Ventricular outflow obstruction can increase end-diastolic pressure and compromise cardiac output in Fontan patients. It is, therefore, necessary during echocardiographic examination to carefully evaluate the semilunar valve, measure the annular and subaortic dimensions, and look for thickness, prolapse, or mobility restrictions. It is also recommended to determine semilunar valve stenosis and regurgitation by using standard Doppler methods. Moreover, the aortic arch must be thoroughly scanned for anatomic narrowing throughout its length [25]. Residual obstruction of the aortic arch can be responsible for arterial hypertension and the consequent increased afterload and ventricular dysfunction.

The lack of a subpulmonary ventricular pump results in non-pulsatile flow in the pulmonary vascular bed and Fontan pathway. As a result, the fluid dynamics of the stenoses will be more like a venous obstruction, and the Doppler flow gradient is frequently low. Therefore, the size of the Fontan connection and of the pulmonary arteries should be carefully examined. Any reduction and enlargement downstream should be taken into consideration and studied by both a continuous and a pulsed wave Doppler (Figure 2). A reduction in the variability of flow and a mean gradient as low as 3 mmHg can reflect a significant obstruction.

### 2.5. Fenestration Flow

Sometimes, at the time of Fontan pathway creation, fenestrations are purposefully kept in place to increase cardiac output. The flow across these fenestrations should be recorded, and the mean pressure gradient should be estimated by continuous wave Doppler echocardiography. In the absence of stenoses, the pressure in the Fontan channel is equivalent to the pulmonary arterial pressure, while the pressure in the functional left atrium will be equivalent to the pulmonary venous pressure in the absence of pulmonary vein stenosis. As a result, the mean pressure gradient across a fenestration will reflect the transpulmonary gradient. A haemodynamically stable Fontan circulation is associated with a transpulmonary gradient of 5 to 8 mmHg.

### 2.6. Collateral Flow

Venovenous collaterals can develop gradually after Fontan surgery and be responsible for significant desaturation [25]. Sometimes they can be suspected from an abnormal low-velocity continuous flow at the suprasternal or high parasternal echocardiographic window. Indeed, the rapid opacification of the atrium following an agitated saline contrast injection in an upper arm would support the evidence of these collaterals.

### 2.7. Transoesophageal Echocardiography

In patients with poor acoustic windows, transoesophageal echocardiography (TEE) can provide a more accurate visualisation. In Fontan patients, TEE can be used to evaluate thrombus, AV valve function, ventricular function, and residual Fontan fenestration (Table 1). In order to see the whole conduit from the IVC to the pulmonary artery, it is crucial to evaluate the Fontan pathway from both the medio-oesophageal four-chamber view and the medio-oesophageal bicaval view. The addition of colour allows for the assessment of Fontan fenestration or residual right-to-left shunts; a more accurate assessment can be obtained by an agitated saline contrast injection in the systemic circulation below the diaphragm. TEE can also be useful in monitoring the interventional devices closure or creation of Fontan fenestration [26].

## 3. Cardiovascular Magnetic Resonance in Fontan Circulation

CMR is an advanced cardiac imaging technique that allows for study of the morphology, haemodynamics, and tissue characterisation of the heart without the use of ionising radiations [27,28]. Images are created thanks to the protons’ net magnetisations, which are modified in strength and directions, while staying in a static magnetic field, by the application of an external radiofrequency together with dynamic gradients [29]. CMR in Fontan patients is considered the gold-standard technique to assess morphology, function, and flows.

CMR can overcome echocardiography’s limitations of suboptimal acoustic windows, operator-dependent measurements, and non-classical cardiac anatomy [10,28,29]. In addition, compared to cardiac computed tomography (CCT) and cardiac catheterisation, CMR is free from ionizing radiation, not increasing the risk of potential future malignant diseases. For this reason, CMR is also offered to children and pregnant women. It is usually performed in awake children from the age of 7–8 years. In younger or non-compliant individuals, CMR needs to be performed under general anaesthesia [30]. Gadolinium-based agent use has been rarely associated with allergic reactions and with new entitles such as nephrogenic systemic fibrosis in patients with severely compromised renal function and brain gadolinium deposition in patients with repeated administrations. The incidence of these events in children has been demonstrated to be the same as or less than in adults [31]. CMR sequences give detailed information on cardiac anatomy, allowing discrimination between different types of Fontan operations used in the past years, and they combine anatomy with data on function and flow dynamics. Lastly, dedicated sequences using a gadolinium-based contrast agent allow for myocardial tissue characterisation, discriminating the presence of myocardial oedema or fibrosis. 

On the other hand, CMR has a long acquisition time that can lead to reduced patient compliance in breath-holding. It has a high cost and limited availability and requires high expertise in the performance and interpretation of the images. Arrhythmias and metallic devices can generate artefacts, but nowadays there are techniques to overcome these limitations. Implantable pacemakers or defibrillators could be a contraindication for CMR, and in these patients these should only be considered in specialised centres [32,33,34]. 

Given the advantages of this diagnostic method, CMR is performed routinely in Fontan patients. In most centres, patients undergo yearly echocardiographic studies and a CMR every five years starting from the age of 8, unless an urgent assessment is necessary [28,35]. 

### 3.1. CMR Protocol

The CMR protocol generally includes the following sequences: dark- and bright-blood single-shot images, balanced steady-state free precession (bSSFP) cine images, flows, a 3D whole heart, and an angiography. Gadolinium agent is administered when needed [36].

CMR studies usually start with dark-blood or bright-blood single-shot imaging in the axial, coronal, and sagittal planes of all of the thorax [37,38,39] These images not only are the base to plan further sequences but also can collect information regarding the intracardiac connections and structures outside the heart, potentially showing unexpected extra-cardiac findings [39].

bSSFP cine images give information on anatomy and function; these are usually breath-hold acquisitions, preferably at end-expiration, but young children may better tolerate the acquisition at end-inspiration. In the non-compliant cases, the bSSFP cine images could be acquired during free breathing, with some decrease in spatial resolution. 

A transaxial cine bSSFP stack, which consists of transaxial slices covering all the vertical length of the heart in the thorax, gives general information regarding heart function and intracardiac anatomy.

bSSFP cine sequences are also performed in long- and short-axis views: a long-axis view of the inflow and outflow of the dominant ventricle, including the atrioventricular valves; an aortic arch should be acquired in case of previous coarctation and/or repair; outflow tracts; a stack of contiguous slices that covers the dominant and non-dominant ventricles from the base to the apex, which will be used to calculate the volumes and ejection fractions; a modified four-chambers view with a horizontal long axis including the dominant ventricle and the eventual rudimentary together with the atrioventricular valves and atria; a modified two-chambers view in a vertical long-axis that includes the dominant ventricle with its respective atrium, Fontan conduit, pulmonary arteries, superior, and inferior vena cava. The presence of an atrial septal defect (ASD) or a ventricular septal defect (VSD) needs to be assessed carefully in order to exclude restrictions at such levels.

Angiography sequences are realised through gadolinium-based contrast agents. The sequence is acquired when the contrast agent arrives in the vessel of interest. In this way, it is possible to study the Fontan pathway, the great vessels, and the venovenous, arterial venous malformation, and aortopulmonary collaterals. In addition, it helps to clarify the flow direction, potentially discovering stenosis and blockages of the blood pathways. Twist is another angiography protocol based on multiple and continuous acquisitions of the arterial, mixed, venous-phase images during the passage of a contrast agent through the vascular anatomy.

Moreover, 3D whole-heart SSFP allows a comprehensive and detailed evaluation of intracardiac and extra-cardiac morphology; it permits the visualisation of the arterial and venous systems together with their collaterals and the measurements of their vessels’ size. In adolescents and adults, it is acquired during mid-diastole when the heart is still, but, in children, due to the high heart rate, it may be necessary to plan it at the end-systolic phase of the cycle Figure 3.

Flow across a vascular structure can be assessed through phase-contrast imaging (PCI), which can quantify velocities and blood flow volumes. Two-dimensional (2D) PCI remains the clinical standard for flow quantification, and 2D PCI are conventionally acquired in the following planes: aorta at the sinutubular junction, superior vena cava, inferior vena cava, Fontan tunnel (pre- and post-fenestration), right and left pulmonary arteries, and right and left pulmonary veins. It can also be used to assess flow velocity across VSD, Damus–Kaye–Stansel anastomosis, and the aortic coarctation site in order to study the presences of stenosis, which can be deleterious in the context of a Fontan circulation.

A new type of sequence that is gradually affirming itself in CMR scans is 4D flow. This might require longer acquisition times, but with a single free-breathing sequence, reproducible data on the three-directional velocities inside all the cardiac structures is obtained. In addition, 3D velocity encoding allows for the determination of the global and local blood flow characteristics. There are also promising data that correlate flow viscosity with liver fibrosis/oedema [37,40].

At the very end of the scan, between 10 and 20 min from the time of injection of the gadolinium-based contrast agent, the late gadolinium enhancement (LGE) sequences are acquired, allowing for the detection of the presence and extent of myocardial fibrosis. These sequences should be performed in cases of heart failure symptoms or arrhythmias. LGE imaging is the gold standard for assessing myocardial fibrosis, relaying on the different washout kinetics of a gadolinium-based contrast agent in healthy and fibrotic myocardium. In the latter, the higher interstitial volume increases the washout time, letting the myocardium appear brighter when the images are acquired. In Fontan patients, fibrosis can be due to endocardial fibroelastosis, previous open-heart surgery (scarring), or mid-wall fibrosis (myocarditis or ventricular sheer stress) [41,42].

### 3.2. Use of CMR for Identification of Fontan Circuit Complications

The possibility of obtaining detailed information on anatomy, function, and haemodynamics is paramount to identify, rule out, and understand the pathophysiological background of many complications of the Fontan system.

bSSFP images permit the assessment of regional wall motion abnormalities and myocardial systolic impairment. bSSFP images can also calculate the end-diastolic and end-systolic volume of the uni-ventricle or both the main and rudimentary ventricles. The end-diastolic volume index is an important, strong predictor of death and transplant-free survival [9,43].

As already mentioned, atrioventricular valve regurgitation is a frequent complication in Fontan failure. It can be visually detected through bSSFP sequences and quantified via flow sequences. Nevertheless, the regurgitation fraction can also be precisely derived by the difference between the stroke volume of a single functional ventricle and the systemic forward flow [44].

bSSFP can also lead to the suspicion of the ventricle’s in- and outflow obstructions, and in-plane flow sequences can help in identifying accelerations and estimating peak velocity. 

The obstruction and stenosis of pulmonary arteries, systemic veins, and pulmonary veins can also be present and must be excluded every time a CMR is performed, bearing in mind that small flow acceleration can have important consequences in Fontan patients and on the lymphatic circulation and can lead to a further increase in other organs’ venous pressure. Flow imaging presents the flow distribution patterns of caval flows and pulmonary arteries, giving important information in support of potential transcatheter or surgical reinterventions. 

Thromboembolic complications are another frequent and life-threatening complication. Early gadolinium enhancement (EGE) is a sequence developed to assess clotting, as thrombotic avascular structures appear low in signal compared with the surrounding vascular structures. In comparison to LGE, EGE are acquired before in time after gadolinium administration. In patients with AP Fontan, especially in atrial arrhythmias, the presence of a clot should always be suspected and ruled out.

Desaturation is a frequent finding in Fontan patients, and this can be due to different conditions.It can be associated with conduit fenestration leading to a right-to-left shunt at the atrial level. It can be due to pulmonary-to-systemic venous collaterals leading to desaturation and a right-to-left shunt. It can also be due to the presence of an arterial venous malformation at the lung level (for example, as the consequence of a hepatic factor absence); this is usually not characterised by a shunt. The CMR allows fora precise calculation of the contribution that the venovenous collaterals flow to the systemic cardiac output. The identification of a large collateral can lead to a cardiac catheter intervention, if the desaturation is significant [45]. 

Fontan-associated liver disease and lymphatic dysfunction can be investigated through CMR, though these do not enter the standard heart protocol and necessitate ad hoc protocols. However, awareness of the long-term detrimental effects on other systems of a passive Fontan circulation is increasing, and in the future more extensive studies are warranted.

## 4. Cardiac Computed Tomography in Fontan Circulation

Due to ionizing radiation exposure, CCT has been historically overlooked in CHD patients, particularly in those requiring serial imaging evaluation. However, the introduction of new generation scanners and the new strategies adopted to minimise both radiation exposure and acquisition time have significantly increased the role of CCT even in this clinical scenario. The use of dual-source or wide-detector scanners, as well as sucrose oral administration and immobilisation devices in neonates, allows for obtaining good quality images even in the youngest of patients, without the need for sedation or general anaesthesia [46]. The latter should be reserved for extremely non-compliant patients or if longer acquisition times are expected due to the use of older scanners. The risk of serious adverse events is low but not negligible, thus requiring adequately trained staff and appropriate equipment [47]. In order to reduce radiation exposure, a scan should be limited to the region of interest, positioning the heart at the isocenter of the gantry and using the appropriate tube potential and tube current [46]. All these precautions seem sufficient to minimise exposure, so organ shielding is not routinely recommended in the paediatric population due to the potential risk of interference with automated dose-modulation systems [48]. To further reduce the radiation dose, a non-contrast scan should be avoided due to the very limited information provided in this clinical context. Conversely, iodinate contrast is crucial, and the administration protocol should be tailored to answer the clinical question with the smallest amount of contrast, avoiding unnecessary repeated acquisition. The aim would be to acquire the images when the contrast has an optimal concentration in the region of interest, by using faster acquisition protocols and considering the cardiac output, the site of injection, and the injection rate. Particular attention should be paid to eliminating all the air bubbles from the contrast, particularly in these patients, due to the presence of septal defects, fenestration, and collaterals and the consequent potential risk of right-to-left embolization (Figure 4).

In patients with a Fontan anatomy, CT could be the preferred imaging modality when a concomitant evaluation of the coronary anatomy is required or in the presence of cardiac devices (either contraindicated or the source of unpredictable artefacts in CMR) [32,33,34]. This may be particularly relevant when considering the increased risk of pacemaker implantation described in this population [49]. An ECG-gated CT can also accurately provide information about left-ventricular volumes and function with good agreement with CMR measurements, despite the lower temporal resolution [50]. However, this requires a higher dose of radiation and an optimised contrast-acquisition protocol if right-chamber quantification is also needed, to maximise RV opacification and minimise artefacts [50]. The major indication in Fontan patients for CCT remains the evaluation of potential thromboembolic complications, occurring in 1–33% of patients and still representing one of the major causes of morbidity and mortality [51]. Thrombi usually appear as hypodense eccentric areas within a vessel after contrast administration that persist in the delayed phase. Obtaining a homogenous opacification of the pulmonary vessels could be particularly challenging in Fontan patients due to the peculiar anatomy and the consequent different timing of contrast arrival in the left and right systems. This generates a mixture of opacified and non-opacified blood that could generate artefacts and mimic the presence of a thrombus [52]. To avoid this possible pitfall, different protocols have been proposed, such as changing the number of sites of injection or the number and timing of the acquisition phases [53]. In the dual-injection protocol, two veins (from the upper and lower venous systems) are cannulated and used for simultaneous contrast injection. Despite increasing the diagnostic performance, this technique has the obvious limitations of double venous access, which is sometimes difficult to obtain, and the discomfort related to the multiple contrast injections. Single-injection protocols with either a single-but-delayed phase at 75–80 s or a dual-phase acquisition have also been proposed, which demonstrated a greater accuracy when compared to traditional protocols [53]. CCT is also useful in detecting and quantifying stenosis of the conduit due to calcification or intimal hyperplasia and could be used for pre-procedural assessment [54]. Due to its excellent spatial resolution, CCT can also accurately visualise aortopulmonary collaterals, pulmonary arteriovenous malformations, or systemic-to-pulmonary venous collaterals that guide the percutaneous procedures or evaluate the outcomes during follow-up [55]. Fontan patients often suffer from liver disease, a consequence of the chronic, systemic venous hypertension. An abdominal CT early identifies hepatic congestion as a reticular enhancement in the portal venous phase. Focal nodular hyperplasia-like (FNH-like) lesions are also a common finding in Fontan patients. These appear as hyso-hypodense lesions, quickly enhancing in the arterial phase and becoming hypo-hysodense in the portal venous and delayed phases, with possible residual enhancement of the central scar [56]. Hepatocellular carcinoma shares some common imaging features with FNH-like lesions, but the atypical enhancement patterns could represent a red flag for a differential diagnosis [57]. Cardiac CT is, therefore, a valuable alternative to other imaging modalities for biventricular volumes and function evaluation, particularly when dealing with patients with a poor acoustic window, patients unable to lie flat for long time, or patients with cardiac devices. CCT could be recommended for pre-operative assessment, with the possibility of obtaining 3D models that are helpful in surgical planning and training. In addition, CT angiography is the key modality for the evaluation of thromboembolic complications and is helpful in evaluating stent patency as well as patency of the pathway and the presence of collaterals. Beyond cardiovascular examination, CT could be also required for lung evaluation in the case of suspected plastic bronchitis or to assess the degree of hepatic involvement. 

## 5. Cardiac Catheterisation in Fontan

When discussing the role of cardiovascular imaging in the follow-up of patients with a Fontan circulation, it is crucial to also mention cardiac catheterisation. Cardiac catheterisation is an invasive procedure in which a catheter is guided through a blood vessel to the heart, not only allowing for the haemodynamic assessment of the circulation but also determining the cardiovascular anatomy, which permits the possibility of intervening in any eventual complications. Through this catheter, the following parameters can be measured: central venous pressure, cardiac output (CO), pulmonary vascular resistance (PVR), systemic vascular resistance (SVR), and the hepatic venous pressure gradient (HVPG) [58]. Knowledge of these parameters can help to identify pre-clinical failure scenarios or find and address causes of a current failure status. However, the scientific literature shows that adverse outcomes correlate when pressures are severely abnormal. Moreover, cardiac catheterisation can detect obstructive lesions in the Fontan circuit and coronary circulation and the subsequent interventions by using ballooning or stenting procedures on the aforementioned (obstructive lesions); individualising venovenous (VV), pulmonary arteriovenous, and aortopulmonary collaterals; closing them with apposite coils; then assessing the patency of Fontan fenestrations; eventually closing these with proper devices; determining the function of the systemic ventricle; and, finally, identifying the regurgitation or stenosis of the aortic and systemic atrioventricular valves [59]. Although these are crucial advantages, cardiac catheterisation is an invasive investigation performed under general anaesthesia for paediatric patients or in non-compliant patients, and patients are also exposed to ionising radiation. In addition, general anaesthesia could be an important confounder in pressure measurements and can cause mistakes. To produce reliable and reproducible data, care should be taken in recording all measurements at the same physiological state/phase of mechanical ventilation and rhythm. Indeed, the advantages and drawbacks of cardiac catheterisation have led to the various practices across institutions having no unanimous consensus on the clinical utility of routine investigations. The American Heart Association’s scientific statement on the evaluation and management of children and adults with a Fontan circulation suggests performing this exam in paediatric patients only if it is clinically necessary and during adulthood only every 10 years unless an urgent indication presents itself. Clinical scenarios can lead to cardiac catheterisation and reduced exercise capacity, new hypoxia onset, arrhythmias, protein-losing enteropathy, plastic bronchitis, pleural effusion, and liver failure, which can be all signs of Fontan failure [4].

After Fontan completion, ventricular volume corresponds to superior vena cava (SVC) return, which is connected to the pulmonary arteries, and the systemic venous return from the IVC. The conduit could be initially fenestrated, creating a right-to-left shunt, improving the ventricular preload by reducing the total systemic venous vascular resistance, and improving the overall cardiac output. Understanding this pathway and physiology is fundamental for interpreting the haemodynamic data and where these are collected during this examination. Indeed, pressures are sampled in all the parts of the circuit together with a pull-back gradient across the eventual reconstructed aortic arch, to exclude any obstruction that should be addressed by transcatheter balloon angioplasty if the rise is equal or more than 10 mmHg. The transpulmonary gradient is the most critical pressure measurement in the whole circuit assessment, which equals the difference between the Fontan pressure and the pulmonary venous atrial mean pressure. The expected value is considered 3 mmHg; while it can reach 5 mmHg, which is already high, 20 mmHg is strongly associated with poor outcomes [59,60]. Saturations are also sampled, which can suggest the presence of arterial collaterals to the lung, when there is an important jump between the SVC and the distal pulmonary arteries or the potential presence of pulmonary arteriovenous fistulae in the case of low saturations in the pulmonary veins. As mentioned above, cardiac output, flows, and resistances can be measured and give important information regarding the circuit. CO is known to be low at rest and to increase slowly during exercise. However, patients with a higher CO seem to experience a higher mortality. A high CO is considered secondary to Fontan-associated liver disease for inducing a reduction in SVR. In addition, pulmonary blood flow augmentation and the resultant exaggerated rise in PVR have been associated with reduced exercise capacity, impaired quality of life, excess liver stiffness, elevated N-terminal pro-brain natriuretic peptide (NT-proBNP), and worsened renal function. Finally, HVPG, measured by the difference between the hepatic vein wedge pressure and free hepatic venous pressure, can identify patients with portal hypertension. When the pressure is higher than 5 mmHg, portal hypertension is associated with adverse outcomes, while more than 12 mmHg relates to worse outcomes [58].

## 6. Discussion

It has been almost 54 years since Fontan and his team started to perform the palliative procedure, now named after him, which revolutionised the natural history of patients with a single-ventricle physiology [61]. The original operation combined an anastomosis between the superior vena cava and the distal end of the right pulmonary artery with an end-to-end anastomosis of the right atrial appendage to the proximal end of the right pulmonary artery by means of an aortic valve homograft. Since this first description, the technique has been extensively modified to lead to the contemporary TCPC [62]. These technical evolutions reflect parallel improvements in the understanding of univentricular heart circulation, and cardiovascular imaging plays a role in this sense [63]. 

When discussing the role of multimodality imaging in patients with Fontan circulation, it is necessary to mention that such imaging is fundamental in all the stages before TCPC, guiding clinicians to select good candidates for the Fontan pathway and identifying the right timing for the different steps [64,65]. Indeed, although the Fontan operation has significantly improved the survival in patients with single-ventricle physiologies [66], long-term morbidity and mortality are still important concerns, and several cardiac and extracardiac complications can occur [67,68]. Thus, the selection of candidates is the first step to increase the chances of survival after Fontan completion. 

In the follow-up of patients with TCPC, the choice of imaging modalities will vary according to the patients’ factors (i.e., age), availability of imaging technologies, and clinicians’ skills. In any case, an at-least-annual clinical and imaging surveillance in specialised adult congenital heart disease centres, involving different professional figures, is mandatory. Indeed, imaging interpretation in these patients requires familiarity with a wide range of anatomical abnormalities and with different surgical techniques [69].

Two-dimensional echocardiography remains the first-line imaging to assess the patency of the Fontan circuit, single-ventricle and valves’ function, and aortic outflow obstruction [10]. The use of deformation imaging and 3D echocardiography in Fontan patients is still not fully integrated in routine clinical practice, but data are emerging about their potential role in the evaluation of single-ventricle performance [70,71], valve morphology and function [72], and to guide interventional procedures [73] in specialised centres. Echocardiography should be performed at least annually and, considering that patients are often the best controls of themselves, it is useful to compare their functional parameters over the years. In some cases, poor acoustic windows and complex anatomies can make the acquisition of parameters that are essential for decision making difficult. Thus, over the last several decades, the use of TEE, CMR, and CT in addition to transthoracic echocardiography has significantly increased. Several combinations are possible, all with the final goal of answering a specific clinical question with the least risk to the patient [74].

When available, CMR is usually considered the more appropriate choice for further assessment, since it avoids iodate contrast and radiation. It is usually reasonable to perform CMR every 3–5 years, starting from the age of 8, unless an urgent assessment is necessary [35], and a baseline assessment is recommended at the time of transition from paediatric to ACHD programmes [75]. This technique can accurately quantify the volume and EF of the dominant ventricle and AV valve regurgitation; it gives accurate information about the patency of the pathways and the aortic arch, and it images collaterals and provides tissue characterisation and feasible 3D reconstructions [74,75]. The long scan time, the need for general anaesthesia for non-cooperative patients, and artefacts due to implanted devices are still the main limitations of this technique, but the development of faster sequences and 3D and 4D flow analyses have the potential to overcome these limitations in the near future [37,40]. 

Lastly, CCT is a valid alternative to CMR thanks to its high spatial and temporal resolution and rapid acquisition time, making this imaging modality ideal for acute situations and a powerful tool for the comprehensive anatomical evaluation of cardiac and extracardiac vascular structures and of vascular stents and to guide surgical or percutaneous interventions [38,75]. Other than in an acute setting, it is reasonable to perform it with a frequency similar to CMR, but well-defined recommendations are not available at the moment. The main drawbacks are radiation exposure and the use of iodine-based contrast agents, but low-dose CCT protocols are currently available thanks to recent advancements in scanner technology and reconstruction methods [46,47].

## 7. Conclusions

The evolution of cardiovascular imaging in the assessment of a Fontan circulation has been characterised by a shift from a singular technique to an integrated multimodality approach. Imaging follow-up should be performed in specialised centres and requires the integration of different professional figures. 

Imaging guidelines help clinicians identify the best technique for a specific clinical question, highlighting the advantages and drawbacks of different cardiovascular imaging modalities. However, physicians should keep in mind that a customised approach may be preferable, to achieve an evaluation as accurate as possible for each patient, to guide their therapy and clinical decisions.

## Figures and Tables

**Figure 1 children-09-01875-f001:**
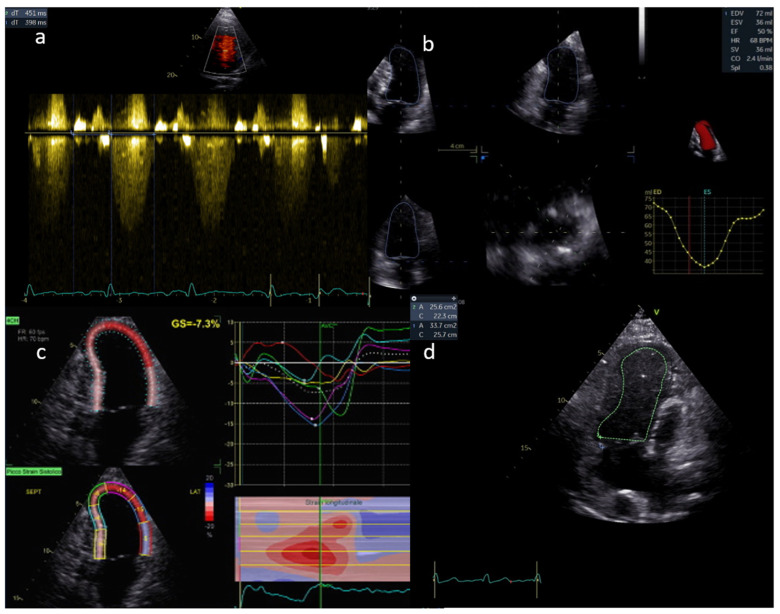
Ventricle function is assessed in **a**, **b**, **c** and **d** by AV S/D, 3D EF, LS, and FAC, respectively.

**Figure 2 children-09-01875-f002:**
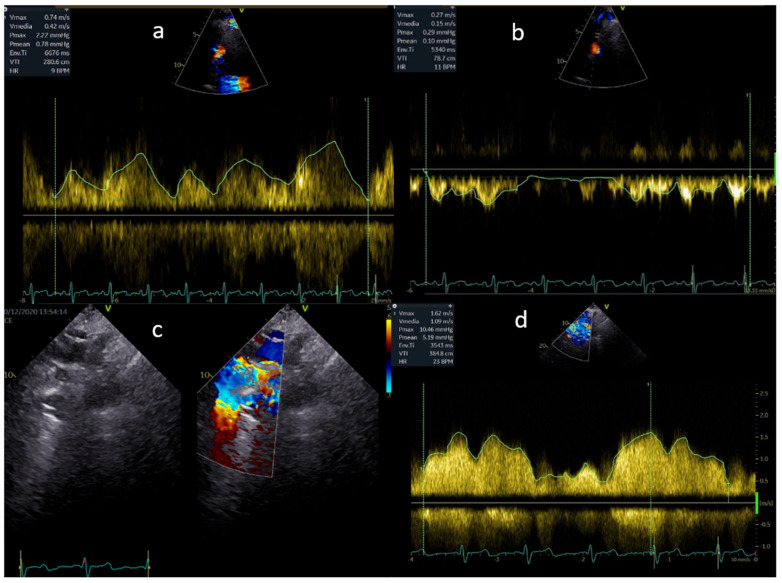
(**a**,**b**) Normal flow in the Fontan pathway; (**c**,**d**) mild flow acceleration through a stented extracardiac conduit.

**Figure 3 children-09-01875-f003:**
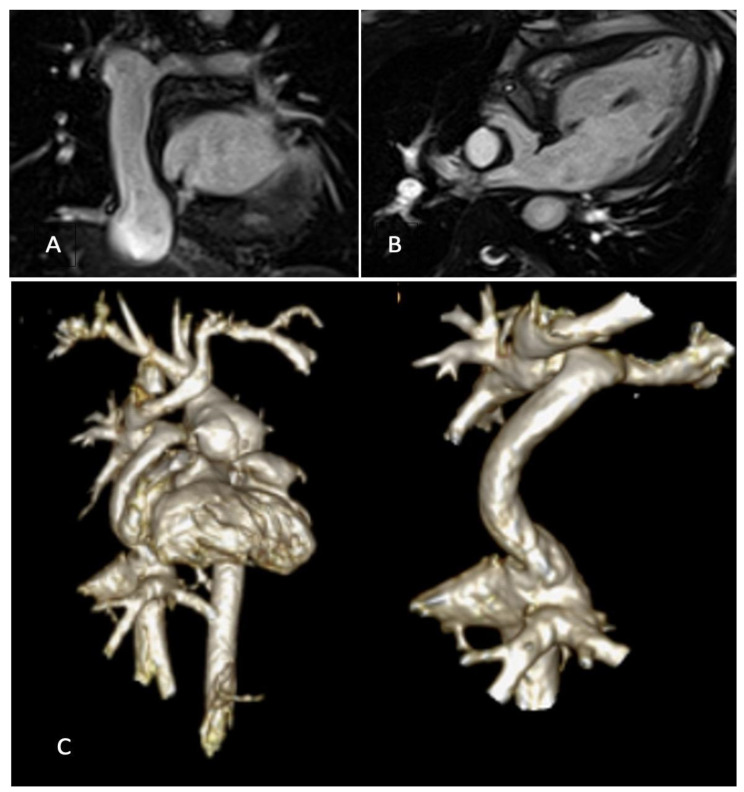
(**A**) bSSFP image of Fontan conduit; (**B**) modified bSSFP four-chamber view with small VSD in the IVS; (**C**) 3D reconstruction of Fontan circuit.

**Figure 4 children-09-01875-f004:**
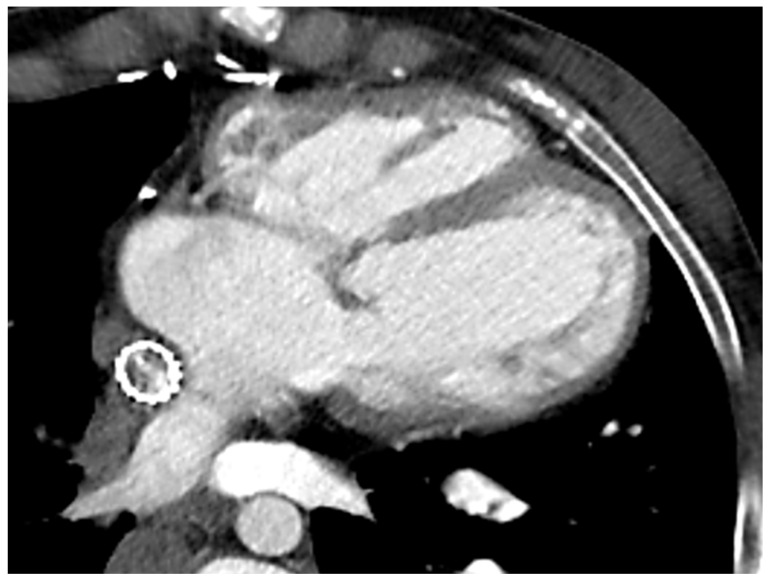
Cardiac CT in Fontan patient with stenting of the extracardiac conduit.

**Table 1 children-09-01875-t001:** Fontan patients’ echocardiographic protocol.

Views Acquisition	Image
Subcostal	Visceral situs Cardiac position and orientation Atrio–ventricular and ventricular–arterial connection ASD and VSD assessment IVC Fontan connection Fenestration flow
Apical	Single-ventricle systolic function: o LV: Biplane Simpson o RV: FAC o Both: 3D EF, dP/dT, AV S/D ratio, STE Single-ventricle diastolic function: o Pulmonary venous atrial reversal flow o E-wave deceleration time AV valve morphology and function: o Regurgitation: § Qualitative assessment § Semi-quantitative assessment: vena contracta width, density, and shape of continuous doppler signal of regurgitant jet o Stenosis: § Mean and peak transvalve gradient · Outflow tract assessment · Semilunar valve morphology and function: o Regurgitation: § Qualitative assessment § Semi-quantitative assessment: vena contracta width o Stenosis: § Mean and peak transvalve gradient
Parasternal	VSD assessment AV valve morphology and function (see apical view) Outflow tract assessment Semilunar valve morphology and function (see apical view)
Suprasternal	Aortic arch obstruction Fontan connection and pulmonary artery flow assessment Collaterals visualisation

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
