# Peer review of "Role of Cardiovascular Imaging in the Follow-Up of Patients with Fontan Circulation"

_children, 2022, doi:10.3390/children9121875_

Round 1

Reviewer 1 Report

Thank you for reviewing this manuscript. The authors discussed the rare congenital heart disease imagining techniques. They summarize and propose surveillance imaging testing to follow-up care for patients who suffer from Fontan circulation. The manuscript is well-documented and gives a review of its scope. 

The weakness of this manuscript is that it doesn't contain any imaging guidelines for the follow-up or the author's recommendations for any guidelines.  After giving any recommendations in the conclusion section, this manuscript will be publishable. 

Author Response

Dear Reviewer,

Thank you very much for your comments. Inside the conclusion section, we have now summarised the indication/guidelines for each cardiovascular imaging method. The manuscript has been reviewed by a native legate speaker in detail.

Please see the final version attached for details.

Reviewer 2 Report

In this manuscript, the authors present a review of multimodal imaging of the cardiovascular system in patients with Fontan circulation.

I have two main points of critique:

(1) "Segmental analysis of CHDs": on page 3, lines 120-122, you write that "The third arrangement, the situs ambiguous, is characterized by a lack of lateralization and abnormal symmetricity of body organs." and that "Since the echocardiographic evaluation of atrial situs can be difficult, abdominal and thoracic situs is generally used as a guide to define the atrial situs."

Comment: Please note that the anomalies of the viscero-atrial situs, which you call "situs ambiguous", in reality do not show bilateral symmetry of body organs. They only show a tendency for bilaterally symmetric arrangements of their inner organs. Thus, they do not completely lack organ lateralization. Moreover, the term "situs ambiguous" is not very helpful. Today, any ambiguity about the visceral situs can be removed by careful multimodal image analysis of the morphology and topography of the thoracic and abdominal organs. Pediatric cardiologists usually distinguish two syndromes of "bilateral symmetry" based on the morphology of the atrial appendages. These are the left and right isomerisms of the atrial appendages. I agree with the authors that the identification of the morphological identity of the atrial appendages, by echocardiography only, is frequently not possible. Unfortunately, however, there is no 100% correlation between thoracic or abdominal features of visceral situs (e.g. (e.g. branching pattern of the bronchial tree or asplenia, polysplenia) and the atrial situs. I miss reference to these facts, which can be found in a huge number of review articles dealing with the sequential segmental analysis of CHDs.

(2) Cardiovascular Magnetic Resonance Imaging: On page 8, line 292-293, you write that "Gadolinium-based agent use in children has been demonstrated to be safe as in the adult population."

Comment: In your manuscript section on Cardiac Computed Tomography, you have nicely discussed the risks of this imaging modality. I, therefore, wonder why you did not refer, in the section on Cardiac MRI, to the current discussion about the safety of Gadolinium-based contrast agents. For reference see for example: Do et al. 2020. Gadolinium-based contrast agent use, their safety, and practice evolution. Kidney360 1(6):561-568.

Further comments: I have found several typos: line 345, arteriorvenous; line 355, sued; line 357, contest; line 376, pato-physiological; line 391, perfumed. Please correct the typos.

Author Response

Dear Reviewer,

Thank you so much for your comments and revision.

  1. The manuscript has been reviewed by a native language speaker. 
  2. We have acknowledged your comment on the echo method and modified it accordingly.
  3. We have inserted one line mentioning the potential adverse reactions of gadolinium-based contrast using your suggested reference,
  4. We have corrected the spelling mistakes that you have promptly identified.

Please see the final version attached with the modifications.
